

# Comparative analysis of the ribosomal DNA repeat unit (rDNA) of *Perna viridis* (Linnaeus, 1758) and *Perna canaliculus* (Gmelin, 1791)

Zhansheng Guo, Leng Han, Zhenlin Liang and Xuguang Hou

Shandong University at Weihai, Weihai, China

## ABSTRACT

*Perna viridis* and *P. canaliculus* are economically and ecologically important species of shellfish. In this study, the complete ribosomal DNA (rDNA) unit sequences of these species were determined for the first time. The gene order, 18S rRNA–internal transcribed spacer (ITS) 1–5.8S rRNA–ITS2–28S rRNA–intergenic spacer (IGS), was similar to that observed in other eukaryotes. The lengths of the *P. viridis* and *P. canaliculus* rDNA sequences ranged from 8,432 to 8,616 bp and from 7,597 to 7,610 bp, respectively, this variability was mainly attributable to the IGS region. The putative transcription termination site and initiation site were confirmed. *Perna viridis* and *P. canaliculus* rDNA contained two (length: 93 and 40 bp) and one (length: 131 bp) repeat motifs, respectively. Individual intra-species differences mainly involved the copy number of repeat units. In *P. viridis*, three cytosine-guanine (CpG) sites with sizes of 440, 1,075 and 537 bp were found to cover nearly the entire IGS sequence, whereas in *P. canaliculus*, two CpG islands with sizes of 361 and 484 bp were identified. The phylogenetic trees constructed with maximum likelihood and neighbour-joining methods and based on ITS sequences were identical and included three major clusters. Species of the same genus were easily clustered together.

## INTRODUCTION

The family Mytilidae comprises a diverse group of bivalves that are broadly distributed in marine environments (*Distel, 2000*). This family includes mussels of the genus *Perna*, which encompasses three currently recognised species of intertidal mussels: *Perna viridis*, *P. canaliculus* and *P. perna*. All three species are considered economically, ecologically and environmentally important (*Wood et al., 2007*). The green mussel *P. viridis* is a warm water bivalve species distributed along the coasts and estuaries of the Asia-Pacific region. This mollusc species is highly prized both as a food source and an important aquaculture component in southeast Asia (*Tan & Ransangan, 2017*; *Wang et al., 2018*). The green-lipped mussel *P. canaliculus* is endemic to New Zealand, and its range extends from the warm temperate northern waters (35.5°S) to the cold temperate waters south of Stewart Island (47°S) (*Wood et al., 2007*). The aquaculture industry surrounding the

Corresponding author
Xuguang Hou,
houxuguang@sdu.edu.cn

green-lipped mussel has expanded rapidly and now represents three quarters of all aquaculture exports (by value) from New Zealand (*Ibarrola, Hilton & Ragg, 2017*).

To date, molecular phylogenetic studies of *Perna* species have been based mainly on mitochondrial cytochrome oxidase I (COI, 623 bp in length) and ribosomal internal transcribed spacer (ITS) (ITS1 & 2, 711 bp) sequence data (*Wood et al., 2007*; *Cunha et al., 2014*; *Gardner et al., 2016*). Although COI and ITS (ITS1 and ITS2) sequences can be used as molecular markers to distinguish *Perna* species (*Wood et al., 2007*), these data do not supply sufficient information for *Perna* classification based on geographical identification or among strains (or even individuals). Within a given locus, short amplicon sequences may include less genetic variation than longer amplicons, which would reduce the ability to distinguish closely related species (*Heeger et al., 2018*). Therefore, this study aimed to identify a sufficiently informative molecular marker useful for inter-species and intra-species phylogenetic analyses of *Perna* species.

In eukaryotes, nuclear ribosomal DNA (rDNA) is usually organised in long tandem repeats. These repeats are arranged head-to-tail in large rDNA clusters to form nucleolar organising regions in chromosomes (*Dyomin et al., 2016*). Each repeated transcribed unit of rDNA comprises a coding region (18S, 5.8S and 28S rRNA genes) separated by two internal transcribed spacers (ITS1 and ITS2). Moreover, each transcribed unit is separated by an intergenic spacer (IGS) which can be subdivided into external transcribed spacers (ETSs) and the non-transcribed spacer (NTS). The coding regions are highly conserved between organisms, and have been selected as genetic markers for higher-level relationships within the molluscan (*Distel, 2000*; *Passamaneck, Schander & Halanych, 2004*; *Combosch & Giribet, 2016*). Whereas non-transcribed regions (ITS1, ITS2 and IGS) exhibit high structural variability (*Li et al., 2016*; *Huang et al., 2017*; *Guo et al., 2018*). The IGS region often contains repeat fragments, promoter and enhancer sites. Both the IGS and its embedded promoters can evolve more rapidly than other parts of the ribosomal repeat unit, which can lead to considerable differences in the sequences between related species and considerable intra-individual variations in length (*Huang et al., 2017*). The IGS region was shown to be more variable and phylogenetically informative than the ITS (*Zhao et al., 2011*). The availability of a complete rDNA sequence would provide researchers with various options, depending on the molecular variability. For *Perna* species, however, the NCBI database only contains partial rDNA sequences (18S and ITS) and a complete rDNA unit has never been reported.

In the present study, the complete sequences of nrDNA in *P. viridis* and *P. canaliculus* are reported for the first time, and the molecular features of various rDNA components are characterised, especially IGS. The phylogenetic relationships among members of the family Mytilidae are revealed based on ITS sequences. Comparative analyses are performed with several known complete rDNA sequences to clarify the rDNA relationships among these molluscs and other eukaryotes.

# MATERIALS AND METHODS

## Mussel material and DNA extraction

*Perna viridis* samples were collected offshore of Leizhou Bay (20.968614°N, 110.325743°E) in China, and *P. canaliculus* samples were collected from Pegasus Bay (43.312595°S,

**Table 1 Primers used in the present paper.**

| Amplified region | Primers | Sequences (5′-3′) | Length (bp) |
|---|---|---|---|
| 18S | 18S-ar | CTTTCAAATGTCTGCCCTAT | 1,505 |
| | 18S-br | TTCACCTACGGAAACCTTGT | |
| 18S-ITS-28S | ITS-ar1 | AGGGACAAGTGGCGTTTAGC | 1,573 |
| | ITS-ar2 | TCGTAACAAGGTTTCCGTAG | 1,162 |
| | ITS-br | TTACCTCTAAGCGGTTTAC | |
| 28S1 | 28S1-ar | TTAGAGGTAAACGGGTGGAT | 1,093 |
| | 28S1-br | AGTTGATTCGGCAGGTGAG | |
| 28S2 | 28S2-ar | GACGAAACGACCTCAACCTA | 1,957 |
| | 28S2-br | AATGATAGGATGAGCCGACA | |
| 28S-IGS-18S | IGS-ar1 | GGGATAACTGGCTTGTGGCA | 2,486 |
| | IGS-br1 | TGGATGTGGTAGCCGTTTCT | |
| | IGS-ar2 | GGATAACTGGCTTGTGGCA | 3,497 |
| | IGS-br2 | CTGCCTTCCTTGGATGTGG | |

172.854629°E) in New Zealand. Three individuals from each species were used in this study. Total genomic DNA was extracted from muscle tissue using the TIANamp Marine Animals DNA Kit (TianGen Biotech Co., Ltd, Beijing, China) according to the manufacturer's instructions.

## PCR amplification, cloning and sequencing

The rDNA fragments were amplified using the primers listed in Table 1. The primers used to amplify the 18S and 28S rRNA genes were designed based on partial sequences corresponding to the 18S rRNA genes of *P. viridis* (EF613234) and 28S rRNA genes of *Geukensia demissa* (AY145405), respectively. The primers used to amplify the 18S–ITS–28S region were designed from the 3′end of *P. viridis* (EF613234) and 5′end of *G. demissa* (AY145405), while the IGS region were designed based on the 3′end of *G. demissa* (AY145405) and 5′end of *P. viridis* (EF613234). Each PCR had a total volume of 25 µl and contained the following: one µL of DNA template, 0.5 µL of each primer (10 µmol/L), 12.5 µL of 2 × TransTaq PCR SuperMix and 10.5 µL of $H_2O$. The reactions were initially denatured at 95 °C for 3 min and subjected to 35 cycles of denaturing at 95 °C for 45 s; annealing at 52 °C for 18S and 28S rRNA, 50 °C for 18S–ITS–28S and 56.5 °C for 28S–IGS–18S for 30 s, and extension at 72 °C for 1–3.5 min (1 kb/min). Finally, all reactions were subjected to a final extension at 72 °C for 7 min. The PCR products were electrophoresed on 0.8% agarose gels, extracted and purified. The purified PCR products were cloned using the pUCm-T Vector Cloning Kit and SanPrep Column Plasmid MiniPreps Kit (Sangon Biotech Co., Ltd., Shanghai, China) according to the manufacturer's instructions. Three positive recombinant plasmid colonies corresponding to each amplified region were picked, cultivated and sequenced by Beijing Ruiboxingke Biotechnology Co., Ltd. (Beijing, China).

## Sequence analysis

The complete rDNA unit was assembled using DNAMAN software (*Kaukonen et al., 2000*), and both sequences have been deposited in the GenBank database (accession

numbers: MK419104–MK419109). The boundaries of each region were confirmed using the NCBI database and BLAST software (*Johnson et al., 2008*). Multiple sequence alignment was performed using T-Coffee software (*Notredame, Higgins & Heringa, 2000*) with manual adjustments. The general molecular features of *P. viridis* and *P. canaliculus* rDNA were calculated using MEGA 7.0 (*Kumar, Stecher & Tamura, 2016*). Sub-repeat (SR) fragments and inverted repeats of the IGS region were analysed with the Tandem Repeats Finder (*Benson, 1999*) and Unipro UGENE (*Okonechnikov, Golosova & Fursov, 2012*), respectively. Predictions of putative transcription initiation sites (TIS) and transcription termination sites (TTSs) in the IGS regions were based on a comparative analysis of the sequences with data for various marine animal species from the literature (*Ki, Kim & Lee, 2009*; *Luchetti, Scanabissi & Mantovani, 2006*; *Chae et al., 2018*; *Guo et al., 2018*). Cytosine-guanine (CpG) islands were identified using CpGPlot (*Polanco & De La Vega, 1994*). The sequence identity was confirmed using DNAMAN software. The genetic distance between *P. viridis* and *P. canaliculus* was calculated with MEGA 7.0 software according to a Kimura two-parameter model. Nucleotide diversity ($\pi$) was calculated using the software DnaSP 5.10 (*Librado & Rozas, 2009*).

For phylogenetic reconstruction, the ITS sequences of *P. viridis* and *P. canaliculus* were first determined. Next, nine complete or near-complete sequences from the following Mytilidae species were obtained from the GenBank database: *Mytilus edulis* (AY695798), *M. unguiculatus* (former *M. coruscus*) (MK157201), *M. galloprovincialis* (JX081670), *M. trossulus* (JX081669), *Aulacomya maoriana* (former *Aulacomya atra maoriana*) (DQ924557), *P. perna* southern Indian population (former *P. indica*) (JQ622200), *P. perna* Omani population (KC692037), *P. perna* Mauritanian population (former *P. picta*) (DQ924548) and *Modiolus rectus* (EF035114). Maximum likelihood (ML) and neighbour-joining (NJ) phylogenetic trees based on ITS sequences were constructed with 1,000 bootstrap replicates using MEGA 7.0 software and the GTR+G nucleotide substitution and Kimura two-parameter models, respectively. *Crassostrea ariakensis* (EU252081) was selected as the outgroup.

## RESULTS

### Complete rDNA sequences of *P. viridis* and *P. canaliculus*

The complete rDNA units of *P. viridis* and *P. canaliculus* were determined after sequencing, comparison and assembly. Figure 1 presents a scheme of the structural organisation of the rDNA repeat unit, which includes the following elements in order: 18S–ITS1–5.8S–ITS2–28S–IGS. The complete rDNA units of *P. viridis* and *P. canaliculus* varied from 8,432 to 8,616 bp and from 7,597 to 7,609 bp, respectively.

The length, GC content, pairwise identity and variable sites of each region are shown in Table 2. An analysis of the *P. viridis* and *P. canaliculus* sequences revealed low levels of genetic variation in the 18S, 5.8S and 28S rRNA regions, which had pairwise identities of 99.56%, 100% and 98.94%, respectively. The variable (V) sites in 18S and 28S rRNA made respective contributions of only 0.83% and 1.22%, indicating that the coding regions (18S, 5.8S and 28S rRNA) were even more highly conserved than the complete rDNA unit (6.25%). Differences between *P. viridis* and *P. canaliculus* were observed mainly in

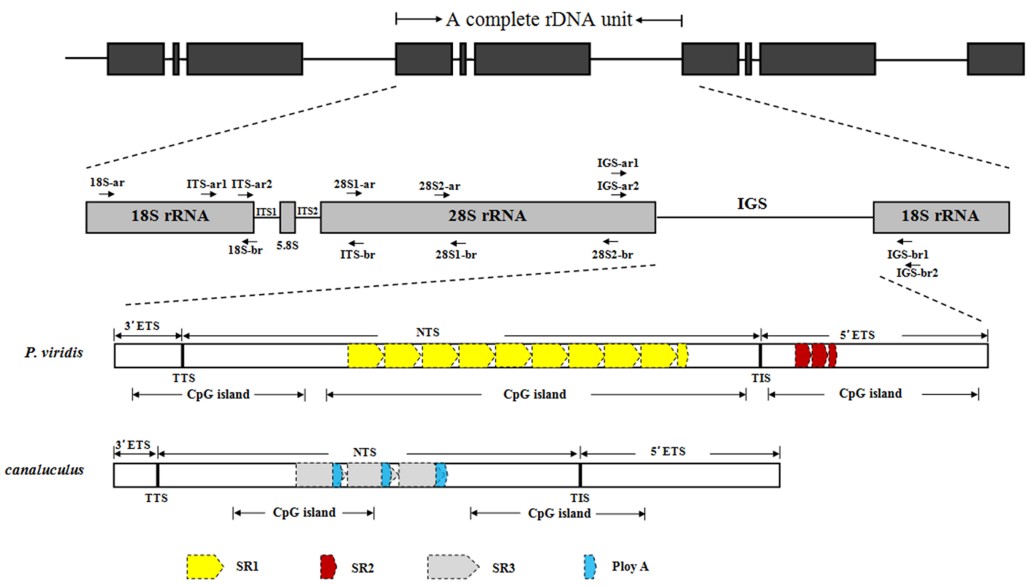

**Figure 1 Structural organisation scheme of nuclear ribosomal DNA (nrDNA) repeat units in *P. viridis* and *P. canaliculus*.** ITS, internal transcribed spacer; IGS, intergenic spacer; ETS, external transcribed spacer; NTS, non-transcribed spacer; TTS, transcription termination site; TIS, transcription initiation site; SR, sub-repeat; CpG island, cytosine-guanine island. The arrows represent primer positions.

**Table 2 Characterization of nuclear ribosomal DNA (rDNA) from *Perna viridis* and *P. canaliculus*.**

| Region | P. viridis | | P. canaliculus | | Alignment length (bp) | Pairwise identity (%) | Pairwise distance | Variable sites (V) |
|---|---|---|---|---|---|---|---|---|
| | Length (bp) | GC content (%) | Length (bp) | GC content (%) | | | | |
| rDNA | 8,432–8,616 | 53.75–53.90 | 7,597–7,610 | 53.02–53.08 | 8,714 | 77.71 | 0.066–0.067 | 546 (6.26%) |
| 18S rRNA | 1,799 | 50.03 | 1,800 | 50.06–50.11 | 1,800 | 99.56 | 0.003–0.007 | 15 (0.83%) |
| ITS1 | 269 | 52.04–53.53 | 304 | 57.89 | 314 | 66.99 | 0.183–0.188 | 45 (14.33%) |
| 5.8S rRNA | 157 | 56.05 | 157 | 56.05 | 157 | 100 | 0.000 | 0 |
| ITS2 | 263 | 51.71–52.09 | 258 | 49.22 | 277 | 67.03 | 0.246–0.252 | 52 (18.77%) |
| 28S rRNA | 3,678 | 54.59–54.65 | 3,679 | 54.61–54.63 | 3,679 | 98.94 | 0.009–0.010 | 45 (1.22%) |
| IGS | 2,267–2,451 | 55.52–55.81 | 1,399–1,411 | 52.18–52.95 | 2,486 | 33.32 | 0.304–0.313 | 389 (15.65%) |

**Note:**
bp, base pair; ITS, internal transcribed spacer; IGS, intergenic spacer; the pairwise distance and identity represent the mean value between *P. viridis* (F1–F3) and *P. canaliculus* (N1–N3), respectively.

non-coding regions (ITS1, ITS2 and IGS). In intra-species comparisons of three individuals per species, the sequence similarities of ITS1 and ITS2 in *P. viridis* were 99.38% and 99.87%, respectively, and these regions differed by only five and one base, respectively. Among *P. canaliculus* individuals, both regions had sequence similarities of 100%. The respective sizes of ITS1 and ITS2 were 269 and 263 bp in *P. viridis* and 304 and 258 bp in *P. canaliculus*. Differences in the lengths and variable sites of the ITS1 and ITS2 regions resulted in relatively low sequence identities (66.99% in ITS1, 67.03% in ITS2).

Comparisons of the IGS sequences at various levels revealed that the highest (91.84% for *P. viridis*, 98.31% for *P. canaliculus*) and lowest similarities (33.32%) were found at the individual and inter-species levels, respectively, and the variable sites between the IGS

**Table 3 Pairwise identities (upper right matrix) and pairwise distances (lower left matrix) of IGS.**

| Sample | F1 | F2 | F3 | N1 | N2 | N3 |
|--------|------|------|--------|--------|--------|--------|
| F1 | | 91.48% | 98.50% | 33.32% | 33.70% | 35.75% |
| F2 | 0.013 | | 92.21% | 33.36% | 32.97% | 33.63% |
| F3 | 0.014 | 0.004 | | 35.90% | 35.39% | 36.24% |
| N1 | 0.327 | 0.323 | 0.320 | | 97.60% | 98.58% |
| N2 | 0.314 | 0.310 | 0.307 | 0.009 | | 98.30% |
| N3 | 0.318 | 0.314 | 0.311 | 0.007 | 0.007 | |

**Note:**
  F1–F3 and N1–N3 represent the three individuals of *P. viridis* and *P. canaliculus*, respectively.

sequences of the two species accounted for 15.65% of all sites in this region (Tables 2 and 3). The sizes of the IGS sequences ranged from 1,399 to 1,411 bp and 2,267 to 2,451 bp in *P. canaliculus* and *P. viridis*, respectively. This result suggested that the IGS sequences of them had large differences in length and a relatively high level of sequence heterogeneity. Due to heterogeneity in the length of IGSs at the individual and inter-species levels of the genus *Perna*, the longest IGS sequence of each species was chosen as a representative to outline the canonical structural organisation of the IGSs from *P. canaliculus* and *P. viridis*. The detailed analysis of the molecular structure of IGS sequences of them contains six distinct regions: TTS, TIS, NTS, ETS, SR and CpG islands.

## Beginning and ending sequences of transcripts, ETS and NTS

A poly (T) tract (5′-TTTTCGTTTGCCTTTTTCGTTCCTTTTTTTTT-3′) was identified in the *P. viridis* IGS region from 171 to 202 bp. The sequence 5′-TTACTTGT-3′ was detected in the *P. canaliculus* IGS region from 120 to 127 bp. Both sequences were considered putative TTSs. Accordingly, the positions of the 3′ETS sequences in the *P. viridis* and *P. canaliculus* 5′IGS regions were determined to correspond to bp 1–202 and bp 1–127, respectively. The sequence 'TTATTATGTGGAGTGGG' was considered a putative TIS for RNA polymerase I, and the +1 position of the TIS was the C/T nucleotide in *P. viridis* and *P. canaliculus*. The 5′ETSs were localised between the putative TIS and the beginning of the 18S rRNA gene and had lengths of 581 bp and 500–503 bp in *P. viridis* and *P. canaliculus*, respectively. Non-transcribed spacers comprised the remainder of IGS. The length, nucleotide diversity, intra- and inter-species sequence identity of 3′ETS, NTS and 5′ETS in *P. viridis* and *P. canaliculus* were shown in Table 4, comparision with 3′ETS and 5′ETS, NTS had lower intra- and inter-species sequence identity and higher nucleotide diversity. The high level of individual, intra- and inter-species divergence within the NTS sequences was attributed to lower functional constraints. In contrast, the nucleotide sequences were more conserved in the ETS region than in the NTSs, despite differences in the length of the former between *P. viridis* and *P. canaliculus*.

## Sub-repeat regions and CpG islands in the IGS sequence

Next, the DNA repeats in the IGS sequences were analysed to explore the IGS genetic structure in greater detail. The identification of the numbers and lengths of the repetitive

**Table 4 The length, nucleotide diversity (π), intra- and inter-species sequence identity of 3′ ETS, NTS and 5′ ETS in *P. viridis* and *P. canaliculus*.**

| Sample | Length (bp) | | | Nucleotide diversity (π) | | | Intra-species identity (%) | | | Inter-species identity (%) | | |
|---|---|---|---|---|---|---|---|---|---|---|---|---|
| | 3′ ETS | NTS | 5′ ETS | 3′ ETS | NTS | 5′ ETS | 3′ ETS | NTS | 5′ ETS | 3′ ETS | NTS | 5′ ETS |
| F1 | 202 | 1484 | 581 | 0.0067 | 0.0144 | 0.0000 | 98.51 | 88.19 | 100 | 71.20 | 54.20 | 84.23 |
| F2 | 199 | 1671 | 581 | | | | | | | | | |
| F3 | 200 | 1485 | 581 | | | | | | | | | |
| N1 | 127 | 784 | 500 | 0.0000 | 0.0113 | 0.0013 | 100 | 97.42 | 99.34 | | | |
| N2 | 127 | 769 | 503 | | | | | | | | | |
| N3 | 127 | 774 | 500 | | | | | | | | | |

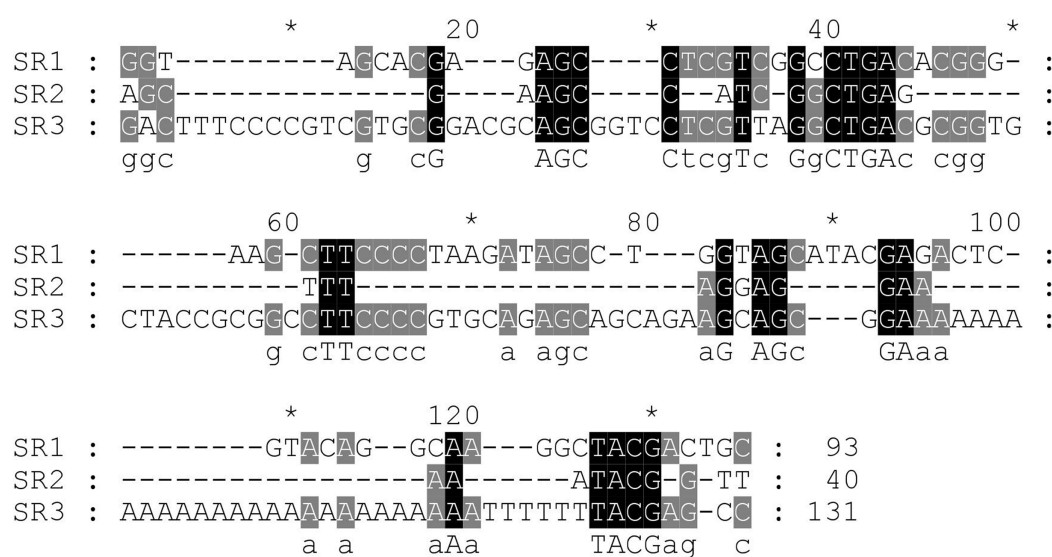

**Figure 2 Alignment of the sub-repeat region of *P. viridis* and *P. canaliculus*.** SR1 (Length: 93 bp) and SR2 (Length: 40 bp) belong to *P. viridis*, and SR3 (Length: 131 bp) belong to *P. canaliculus*. Two nucleotide bases are identical and marked in grey. Three nucleotide bases are identical and marked in black.

sequence motifs occurring in *P. viridis* and *P. canaliculus* enabled the definition of three different SRs with lengths of 40–131 bp (Figs. 1 and 2). A comparison of the IGS regions indicated an extremely high level of sequence diversity within the SR region. The *P. canaliculus* sequence included only one type of SR (length: 131 bp) which was repeated 2.9 times, and three poly (A) tracts in the NTS area. The *P. viridis* sequence included two repeat motifs: SR1 (length: 93 bp), which was repeated 9.3–11.5 times within the NTS, and SR2 (length: 40 bp), which was repeated 2.5 times within the 5′ETS. Differences between individuals of the same species were due to SR copy number variation. No inverted repeats were detected in either *Perna* species.

The GC content of the IGS region was slightly higher in *P. viridis* than in *P. canaliculus* (Table 2). Similarly, the number and lengths of the CpG islands were greater in *P. viridis* than in *P. canaliculus*. In *P. viridis*, three CpG sites were found to cover nearly the

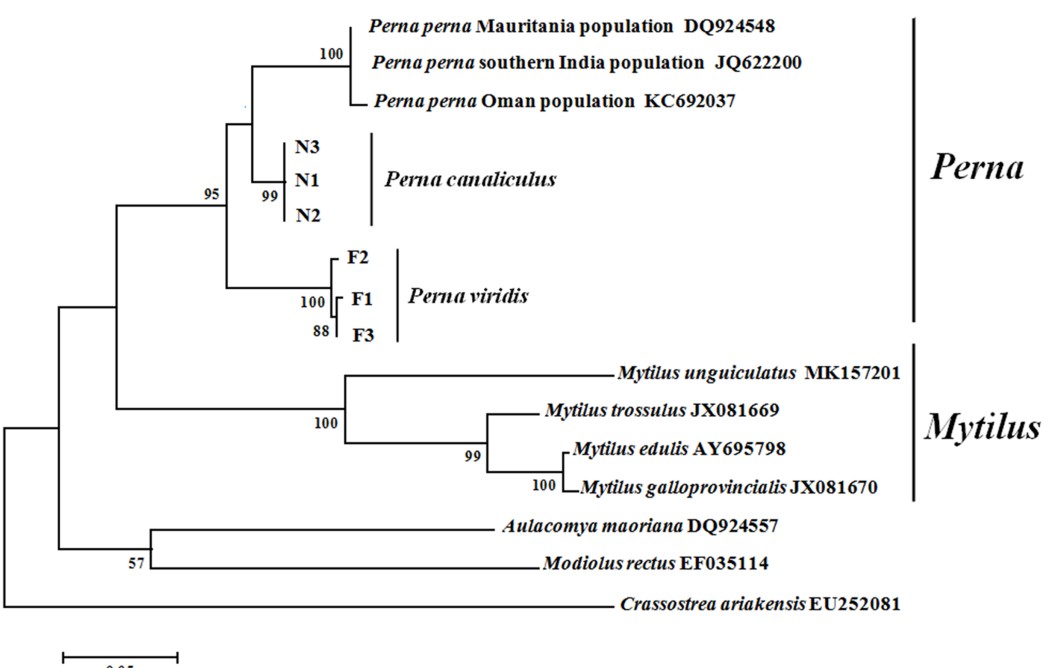

**Figure 3 Phylogenetic trees constructed using maximum likelihood (ML) and neighbour-joining (NJ) methods based on ITS sequences of Mytilidae species.** Numbers around the branches indicate bootstrap support from 1,000 tests. ML and NJ trees are topologically identical. F1–F3 and N1–N3 represent the three individuals of *P. viridis* and *P. canaliculus*, respectively.

entire IGS sequence (lengths: 440, 1,078 and 537 bp). By contrast, the *P. canaliculus* IGS region included two CpG islands located at the NTS and 5′ETS (lengths: 361 and 484 bp).

### Phylogenetic tree analysis based on ITS sequences

Maximum likelihood and NJ phylogenetic trees were constructed based on the ITS sequences of Mytilidae species (Fig. 3). Clustering pattern analysis revealed that the NJ and ML trees were identical, both including three major clusters, and species of the same genus clearly clustered together with relatively high supporting values. For both *P. viridis* and *P. canaliculus*, the individuals were classified into single clusters with high sequence identity. All Mytilidae species included in the phylogenetic analysis belong to the subfamily Mytilinae, except *M. rectus* which belongs to the subfamily Modiolinae. During tree formation, *M. rectus* and *A. maoriana* initially clustered together, and then clustered with *Perna* and *Mytilus* species.

## DISCUSSION

In this study, the complete rDNA units of *P. viridis* and *P. canaliculus* were sequenced for the first time. The rDNA gene orders in these organisms were identical to those in other eukaryotes. Reported rDNA unit lengths vary among taxa, as shown in Table 5, with values of 43 kb (human) (*Gonzalez & Sylvester, 1995*) and 45 kb (mouse) for mammals (*Grozdanov, Georgiev & Karagyozov, 2003*); 13.67 kb for fish (*Cyprinus carpio*) (*Vera et al., 2003*); 12.26 kb (*Brachiola algerae*) (*Belkorchia et al., 2008*), 9.5 kb (*Plasmodiophora*

**Table 5** The rDNA unit length (nrDNA), 18S rDNA, ITS1, 5.8S rDNA, ITS2, 28S rDNA and IGS lengths (bp), and GenBank Accession numbers of eukaryotes used for comparison purposes in this publication.

| Species | nrDNA | 18S rDNA | ITS1 | 5.8S rDNA | ITS2 | 28S rDNA | IGS | GenBank accession No. | Reference |
|---|---|---|---|---|---|---|---|---|---|
| Homo sapiens | 42,999 | 1,871 | 1,095 | 157 | 1,155 | 5,035 | 33,686 | U13369 | Gonzalez & Sylvester (1995) |
| Mus musculus | 45,306 | 1,870 | 1,000 | 157 | 1,088 | 4,730 | 36,462 | BK000964 | Grozdanov, Georgiev & Karagyozov (2003) |
| Gallus gallus | 11,863 | 1,823 | 2,530 | 157 | 733 | 4,441 | 2,179 | KT445934 | Dyomin et al. (2016) |
| Cyprinus carpio | 13,676 | 1,861 | 367 | 158 | 390 | 4,093 | 6,807 | AF133089 and AY260899 | Vera et al. (2003) |
| Entamoeba invadens | 22,481 | 1,962 | 116 | 116 | 78 | 3,188 | 16,700 | AY190083 | Ojha et al. (2013) |
| Brachiola algerae | 12,269 | 1,391 | | 22 | | 2,557 | 7,915 | AM422905 | Belkorchia et al. (2008) |
| Paramphistomum cervi | 8,493–10,221 | 1,994 | 1,293 | 157 | 286 | 4,186 | 577–2,305 | KJ459934–KJ459938 | Zheng et al. (2014) |
| Eurytrema pancreaticum | 8,306–8,310 | 1,996 | 1,103 | 160 | 231 | 3,669 | 1,147–1,151 | KY490000–KY490004 | Su et al. (2018) |
| Plasmodiophora brassicae | 9,513 | 3,105 | 143 | 154 | 165 | 3,611 | 2,332 | / | Niwa et al. (2011) |
| Aurelia sp.1 | 7,731 | 1,814 | 272 | 158 | 278 | 3,606 | 1,603 | EU276014 | Ki, Kim & Lee (2009) |
| Chrysaora pacifca | 8,167 | 1,810 | 246 | 158 | 182 | 3,609 | 2,162 | KY212123 | Chae et al. (2018) |
| Paracyclopina nana | 7,974 | 1,808 | 299 | 157 | 216 | 3,572 | 1,922 | FJ214952 | Ki, Park & Lee (2011) |
| Haliotis discus hannai | 10,668–10,698 | 1,871 | 329 | 160 | 301 | 3,411 | 4,624–4,654 | KY485141–KY485146, KY569413–KY569414 | Guo et al. (2017) |
| Haliotis iris | 9,579–9,706 | 1,858 | 321 | 160 | 296 | 3,412 | 3,560-3,662 | KY933301–KY933305, KY978225–KY978226 | Guo, Hou & Han (2018) |
| Haliotis rubra | 9,881 | 1,858 | 327 | 160 | 297 | 3,413 | 3,854 | MF099780, MF111108, MF106175, MF106174, MF099782, MF113042 | Guo et al. (2018) |
| Oryza sativa | 7,928–8,934 | / | / | / | / | / | / | OSJNOa063K24 and OSJNBb0013K10 | Fujisawa et al. (2006) |
| Stipa spp. | 7,791–8,897 | 1,811 | 221 | 164 | 205-207 | 2,193–3,098 | 3,397 | KY826229–KY826235 | Krawczyk et al (2017) |
| Bangia | 10,976–11,497 | 1,839 | 217–699 | 158 | 398–437 | 3,751 | 4,613 | KP279672–KP279682, KP311305–KP311315 | Xu et al. (2016) |
| Pyropia yezoensis | 13,650–13,654 | 1,834 | 371–372 | 159 | 532–535 | 4,770 | 5,984 | KJ578745–KJ578748, KJ608639–KJ608640 | Li et al. (2016) |
| Marchantia polymorpha | 16,103 | 1,820 | 1,067 | 158 | 381 | 3,468 | 9,209 | AB021684 | Sone et al. (1999) |
| P. viridis | 8,432–8,616 | 1,799 | 269 | 157 | 263 | 3,678 | 2,267–2,451 | MK419104–MK419106 | Present study |
| P. canaliculus | 7,597–7,610 | 1,800 | 304 | 157 | 258 | 3,679 | 1,399–1,411 | MK419107–MK419109 | Present study |

brassicae) (*Niwa et al., 2011*) and 8.3 kb (*Eurytrema pancreaticum*) (*Su et al., 2018*) for parasites; 7.9 kb (*Paracyclopina nana*) (*Ki, Park & Lee, 2011*), 7.7 kb (*Aurelia* sp.1) (*Ki, Kim & Lee, 2009*) and 9.6–10.7 kb (*Haliotis* species) (*Guo et al., 2017, 2018*; *Guo, Hou & Han, 2018*) for marine invertebrates; 7.9–8.9 kb (*Oryza sativa*) (*Fujisawa et al., 2006*) and 8.0–8.9 kb (*Stipa* spp.) (*Krawczyk et al., 2017*) for land plants and 11.76–12.57 kb (*Bangia*) (*Xu et al., 2016*) and 13.65 kb (*Pyropia yezoensis*) (*Li et al., 2016*) for sea algae. Meanwhile, the rDNA lengths of *P. viridis* and *P. canaliculus* were 8.6 and 7.6 kb, respectively.

Table 5 compares each region of the rDNA unit between the two *Perna* species and other eukaryotes. The length of the coding region (18S, 5.8S and 28S rRNA genes) exhibits conservative features. The length of 18S rRNA varies from 1,799 bp to 1,996 bp except for *Plasmodiophora brassicae* (3,105 bp) and *Brachiola algerae* (1,391 bp). The sizes of 5.8S rRNA in most eukaryotes are around 160 bp except for *Entamoeba invadens* (116 bp) and *Brachiola algerae*. In *B. algerae*, the ITS size was estimated at 22 bp and the 5.8S rRNA gene was not detected (*Belkorchia et al., 2008*), while the length of 28S rRNA ranged from 3 to 5 kb. Differences in rDNA lengths among species are mainly attributable to the IGS. According to the database of rDNA unit sequences submitted to GenBank, IGS lengths vary from 0.58 kb in *Paramphistomum cervi* to over 30 kb in mammals.

This study observed variations in the length and sequences of the IGS region in comparisons at both intra- and inter-species levels. Length differences in this region were largely responsible for variations in the entire rDNA unit lengths of *P. viridis* and *P. canaliculus* at both levels and are mainly attributable to differences in indels between individuals. Compared with the entire IGS region and the NTS, the beginning of the 3′ETS and latter half of the 5′ETS sequences were more highly conserved and exhibited greater sequence identity (>65%). All rDNA IGS sequences shared characteristics such as the presence of SR elements. Therefore, heterogeneity in the IGS length can be attributed to duplications or deletions of the SR region, which may be present in different copy numbers in almost all species (*Ambrose & Crease, 2011*; *Huang et al., 2017*). In an intra-species comparison of individuals, length variants were mainly attributable to the SR zone; for example, 9.3–11.5 copies of the repeat motif SR1 were present in *P. viridis* individuals. *Huang et al. (2017)* speculated that indels are restricted to SR-rich regions. Therefore, length polymorphisms in IGS result from concerted evolution, and unequal crossover between SR elements might be a major driving force underlying the evolution of rDNA units (*Ganley & Scott, 1999*; *Stage & Eickbush, 2007*). The repeat fragments were found to be species-specific and rarely adhered to any predictable behaviour or similarities even within a genus. For example, *P. viridis* and *P. canaliculus* exhibited totally different repeat patterns, and this phenomenon has also been observed in other eukaryotes (*Krawczyk et al., 2017*; *Guo et al., 2018*). Possibly, the repeat pattern could be used as a genetic marker for species identification.

The IGS contains several functional elements, including TTS and TIS regions. According to *Ki, Kim & Lee (2009)*; *Ki, Park & Lee, 2011)*, *Guo et al. (2018*; *Guo, Hou & Han, 2018)* and *Chae et al. (2018),* a poly(T) tract on the 5′side of the IGS can be considered a the putative TTS in marine invertebrates. The typical structure of the region

near the beginning of the 5′IGS was also observed in *P. viridis* and *P. canaliculus*, even though the sequences were not totally identical. The core promoter contains TATA and GGGG boxes, which appear to be a general feature of rDNA transcription, and this site is immediately preceded by an AT-rich region, which is commonly found in both plants and animals (*Vera et al., 2003*; *Wang, Zhao & Li, 2003*; *Maggini et al., 2008*; *Krawczyk et al., 2017*). However, the beginnings of the *P. viridis* and *P. canaliculus* IGS genes contained only two and one TATA-box sequences, respectively, and lacked AT-rich regions. Only one previous report described a similar putative TIS sequence (TTATTATGTGGAGTG GG). Specifically, *Krawczyk et al. (2017)*, after an analysis of Poaceae species, also reported that an AT-rich region upstream of the TIS was not ubiquitous. The rDNA genes are widely used to resolve phylogenetic relationships between species at various taxonomic levels. However, few studies have used IGS to identify species or reveal phylogenetic relationships among mussels. A comparison of the genetic distances of rDNA between *P. viridis* and *P. canaliculus* yielded the largest value in the IGS region (0.304–0.313), which suggests that this region could be used to identify species even at a sub-genus level. Given its relatively rapid evolution (compared with other rDNA sequences) and differences in the SR sequences (e.g., copy number, sequence and length) among species, the IGS region may be more suitable than ITS and 18/28S rRNA for reconstructing both inter- and intra-species phylogenetic relationships.

In conclusion, we sequenced the complete rDNA unit of *P. viridis* and *P. canaliculus* for the first time. In both species, the structural organisation of the rDNA unit was similar to those of many other eukaryotes. The unit lengths in *P. viridis* and *P. canaliculus* were 8,432–8,616 bp and 7,597–7,610 bp, respectively, and variations in the length were mainly attributable to the IGS region. We further investigated the characteristics of the IGS and assessed sequence diversity at intra- and inter-species levels. The *Perna* rDNA unit provides a structural model of nuclear rDNA for molecular comparisons, particularly among Mytilidae species. These research discoveries will hopefully pave the way for analyses of mollusc population genetics and evolution.

### Funding
This work was supported by the Shandong Provincial Natural Science Foundation, China (Grant Number ZR2017PEE008). The funders had no role in study design, data collection and analysis, decision to publish, or preparation of the manuscript.

### Grant Disclosures
The following grant information was disclosed by the authors:
Shandong Provincial Natural Science Foundation, China: ZR2017PEE008.

### Competing Interests
The authors declare that they have no competing interests.

## Author Contributions

- Zhansheng Guo conceived and designed the experiments, performed the experiments, analysed the data, prepared figures and/or tables, authored or reviewed drafts of the paper, approved the final draft.
- Leng Han performed the experiments, approved the final draft.
- Zhenlin Liang analysed the data, approved the final draft, polishing the language.
- Xuguang Hou contributed reagents/materials/analysis tools, prepared figures and/or tables, authored or reviewed drafts of the paper, approved the final draft.

## Data Availability

The data are available at GenBank: MK419104–MK419109.

## Supplemental Information

Supplemental information for this article can be found online at http://dx.doi.org/10.7717/peerj.7644#supplemental-information.

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
