# Peer review of "Comparative analysis of the ribosomal DNA repeat unit (rDNA) of Perna viridis (Linnaeus, 1758) and Perna canaliculus (Gmelin, 1791)"

_PeerJ, doi:10.7717/peerj.7644_

## Round 0.1 · original submission · Major Revisions

The document should include more data and analysis as suggested by the reviewers. The improvement of the text and the English language will also be necessary for acceptance.

Reviewer 1 ·

Basic reporting

The MS uses mostly clear, professional English and follows a standard structure.
Some references missing.

Experimental design

The methods employed are adequate and the results seem to be congruent with previously published data from other taxa.
The objectives stated at the end of the introduction were not fully addressed.

Validity of the findings

Please, see general comments

Additional comments

This article describes and analyzes the nucleotide sequences of the major ribosomal DNA repeats in two species of Perna mussels. The MS uses mostly clear, professional English and follows a standard structure. The methods employed are adequate and the results seem to be congruent with previously published data from other taxa. Thus, this is an interesting contribution and merits to be published in Peer J but needs major revision prior acceptation.

The most important issue is that the objectives stated at the end of the introduction were not fully addressed. Although objective 1 conforms to the experimental design used and the results obtained, objectives 2 and 3 are not. To accomplish objective 2 “to assess individual, intraspecies and interspecies diversity in the IGS sequence” many more clones from a single specimen and many more specimens, of different geographical origin, from each species must be studied. Objective 3 “to explore the phylogenetic relationships of P. viridis and P. canaliculus with each and other Mytilidae species” needs wider analyses (i.e., Kartavtsev et al 2018 Rus J Genet, Liu et al 2018 Mol Phylogen Evol). Comparing the 18S and 28S rDNA sequences of those two Perna species with those described for other eukaryotes (Saccharomyces, Drosophila, mammals...) would be a more interesting and feasible objective.

Other issues that must be addressed are

All species names, and their taxonomic status, that appear in the MS must be revised following the World Register of Marine Species (WoRMS, http://www.marinespecies.org/). Some examples
Perna picta is unaccepted the accepted name is Perna perna (Linnaeus, 1758)
Perna indica is unaccepted the accepted name is Perna perna (Linnaeus, 1758)
Aulacomya atra maoriana is unaccepted the accepted name is Aulacomya maoriana (Iredale, 1915)
Mytilus coruscus is unaccepted the accepted name is Mytilus unguiculatus Valenciennes, 1858
Geukensia demissa (Dillwyn, 1817) belongs to subfamily Brachidontinae

The references along the text must be ordered chronologically (i.e. L43-L44)

Introduction
The common name for Perna canaliculus Gmelin 1791 must be checked. The classical common name for this species was green-lipped mussel and I am not sure about the TM label for greenshell mussel. See for instance Wood et al. 2007.
L45: identification instead of classification?
L50: usually? Is it any exception to the rule?
L50-63 paragraph must include some references to other members of the family Mytilidae, as well as to other bivalves, and cite Passamaneck et al. 2004 Mol Phylogen Evol.
The final paragraph of the introduction (L64-68) must be reformulated.

Material and Methods
What’s the rationale for using so many pairs of primers? How long were the amplified fragments?
Considering the structure of the major (45S) rDNA in eukaryotes, was not enough using two pairs of primers, one for the IGS and another one for the coding region?
Taking into account the nucleotide sequences provided as raw data, the primer pair IGS-ar1 and IGS-br-2 must work in both species, was that not the case?
Is something missing in L90 (Fr Three)? Were three different clones for each specimen sequenced?
L90: How were rDNA units assembled? Were consensus sequences for each species determined?
L93-94: Which sequences were used to determine boundaries between regions? Those mentioned in L100-101?
L104-111: Why selecting only those species of Mytilidae? Why using an abalone as outgroup? Why not using other Pteriomorphia bivalves, i.e. Ostreidae? I would suggest removing this part from the MS

Results
The results are not clearly and unambiguously displayed. There are some sentences that belong to discussion.
Figure 1 is not informative enough and must be changed to include the localization of the primers. The nucleotide in which transcription starts and the nucleotide positions (from the consensus sequences for both P. viridis and P. canaliculus) in which all regions (5’ETS, 18S, ITS1, 5.8S, ITS2, 28S, 3’ETS, NTS) start and end must be also signaled.
Maybe a single figure including this information and that provided in Figure 2 would be more adequate.
Were three different clones for each specimen (and each amplicon) sequenced? Were the sequences for each specimen identical or showed differences? Were the six sequences deposited in GenBank (not released) consensus sequences or identical?
Information in table 2 is somewhat misleading. As one of the implicit objectives of the work (L47-49) is the possible use of variation in IGS nucleotide sequences in intra-specific analysis, in my opinion pairwise identities and distances for each species must be also provided.

Discussion
I wouldn’t use the word gene for the IGS.
I suggest focusing the discussion in the comparative analysis of the rDNA unit, including the more conserved coding regions, both between the two Perna species and with respect to other eukaryotes.
L173-185: Instead of centering the paragraph only in the differences in length of the whole repeat, I suggest also compare the “conserved” sequences (18S, 5.8S and 28S)
L209-223: I suggest removing, or drastically reducing, this part from the discussion

Reviewer 2 ·

Basic reporting

-

Experimental design

-

Validity of the findings

-

Additional comments

Even when the paper is valuable because of the results are information that practically does not exist in molluscs. However, they should include more analysis to the sequence data as for example secondary structure analysis and diversity. I also suggest that they analyze a higher number of individuals or clones per individual for each species. My suggestion is that they complete the data and analysis, and they submit again the paper.

Reviewer 3 ·

Basic reporting

The manuscript submitted by Guo et al deals with the sequencing and analysis of the rDNA cistron in two species of Perna shellfish, with special regard to the IGS and ITS region. The analysis describe in detail the IGS sequences, including variation of sub-repeat arrays, and shows a phylogenetic analysis based on the ITS sequence.
While the analysis is interesting and potentially leading to efficient application of IGS as species recognition marker, I think the whole manuscript need improvement of language and analyses before resulting acceptable.

The English language need to be improved: in many instances incorrect/weird sentences and /or words are used. I listed some go them in the General Comments for the authors.
References are sufficient and the background appears complete. Article structure is in line with current standard; tables and figures are well done, self-explicative and complete. Raw data are shared and submitted to the relevant on line repository.
The analysis is rather descriptive and somewhat limited, although envisaging the use of obtained data as a basis for future application. Unfortunately, this remains only a hypothetical use, and no practical tests have been carried out: this makes the analysis a little bit incomplete.

Experimental design

I can’t see any flaw in the experimental methods: data isolation and elaboration are in line with current standard. I may question the use of the Neighbour-Joining for phylogenetic tree reconstruction, as it is greatly outdated and error-prone, but for the limited dataset presented in the analysis could be acceptable.

Validity of the findings

The manuscript is rather descriptive, but this is OK. I think that it could be greatly improved if methods addressing the aim of species recognition and results are presented. I think that that’s could be not a great deal to perform and would demonstrsate the really utility of IGS analysis for Perna species identification.

Additional comments

As said, the manuscript is rather descriptive and, to some extent, limited. I would suggest to authors to improve the level of the analysis to better addressing the aim stated in the background section.
Moreover, I think that language need to be improved; while a commercial editing service of the help of native speaker colleagues are necessary, here I listed a bunch of examples where word/sentences need to be fixed:

line 43: weird notation CoI. it is more generally indicated as COI or cox1;
line 51: "arrayed" should be "arranged";
line 57: "fragments and promoter and enhancer" should be "fragments, promoter and
enhancer";
line 65-66: IGS is not a gene;
line 136: please, motivate why those regions were considered TTSs;
line 140: please, define "common zone". Could it be a shared/conserved region?;
line 147: "better" should be "more"
line 150: IGS is not a gene;
line 157: it is unclear if the difference was attributed to SR copy number or it means that the difference is actually due to SR copy number variation. Please, explicit this point;
line 165: "cluster analysis" should be "clustering pattern analysis" (the fact is that the former is the name of a statistical method which, I assume, has not been used for this purpose);
line 170: I think that "relatively closely related", in this instance, should be simply "more related";
line 174: why "notably"? I don't think there were hints of different expectation;
line 175: avoid the use of terms such as "type of organisms": terms like "taxa" or the same "organisms" is enough descriptive and more scientific.
line 185: please, define "higher evolutionary level". Probably, author want to refer to the fact that IGS appear longer in vertebrates. The sentence "higher evolutionary level" is, technically, non-sensical: I suggest to avoid it.
line 188: base substitutions do not affect sequence length;
line 198: please, define "rules";
line 200: "this phenomenon could" should be changed with "the repeat pattern could";
line 219: please, replace "phylogeny tree" with "phylogenetic tree";
line 227: IGS is not a gene.

---

## Round 0.2 · Minor Revisions

The manuscript must follow the indications of the three referees before being accepted. This is especially necessary to resolve the errors described by referees # 1 and 2 and the quality of the English language mentioned by referee # 3. The suggestion to use green-lipped mussel by referee # 1 must be followed or answered with a strong argument against.

Reviewer 1 ·

Basic reporting

No comment

Experimental design

No comment

Validity of the findings

No comment

Additional comments

PeerJ-36103-v1 Guo et al. 2019
Comparative analysis of the ribosomal DNA repeat unit (rDNA) of Perna viridis and Perna canaliculus

Although the reviewed version of the article addresses most of my previous comments, the MS still needs further revision before being ready for publication.

I am still not sure about the convenience of using a trade mark as the common name of a species, the editor has to make a demission but I suggest using green-lipped mussel instead of greenshell (TM) mussel.

The terms rRNA, rRNA genes and rDNA are incorrectly used along the MS. See for instance, the start of the second paragraph in M&M (L71-73)
The rDNA genes were amplified using the primers listed in Table 1. The primers used to amplify the 18S and 28S rRNA regions were designed based on partial sequences corresponding to the 18S rRNA of P. viridis (EF613234) and 28S rRNA of Geukensia demissa (AY145405), respectively
The rRNA genes (or rDNA fragments) were amplified using the primers listed in Table 1. The primers used to amplify the 18S and 28S rRNA genes (or 18S and 28S rRNA coding regions) were designed based on partial sequences corresponding to the 18S rRNA genes (or 18S rDNA) of P. viridis (EF613234) and 28S rRNA genes (or 28S rDNA) of Geukensia demissa (AY145405), respectively.

The wording of the last paragraph of the Results section (L164-170) must be revised and minor spelling mistakes corrected. Why not include Geukensia demissa (Dillwyn, 1817) ITS sequences (AY621983.1, AY621962.1) in the tree?

The caption of Table 4 (the species were collected with rDNA repeat unit) does not make sense

Many more mistakes appear along the MS, i.e.
L10: P.canaliculus
L14: thisvariability
L17: the copy numbers of repeat units?
L20; Species of the same genus were easily clustered in a single clade?
L38-39: information for classification based on geographical identification or among strains (or even individuals)? Those mussels are already classified, we only assign specimens to a taxon or identify them
And so on.

Reviewer 2 ·

Basic reporting

Revision to the ms. “Comparative analysis of the ribosomal DNA repeat unit (rDNA) of Perna viridis and Perna canaliculus” (#36103) from Guo et al.


Title: I suggest to add the authority after the scientific names.
Lines:
14: Could you change “thisvariability” to “this variability”.
17 and 18: Whereas sizes of the CpG islands are indicated for P. canaliculus, they are not inidicated for P. viridis.
15-17: It would be illustrative if you include the length of the repeats. In the same way the CpG islands can be better described adding the length.
Introduction
37: ITS should read ITS (ITS1 and ITS2).
37: Gardner et al. (2916) says that the length for P. viridis in COI and ITS are 650 pb and 670 pb. respectively. Besides, the data of length reads a little odd there, maybe this data are better in line 36.
38: After “Perna species” include the reference (Wood et al., 2007).
39-41: This may be not true as microsatellites of sizes 1-6 pb can effectively distinguish species and individuals based in the number of repeats.
41-42: This is a contradiction as you say in lines 37 and 38 that ITS does not supply enough information for intraspecies comparisons and you say you are looking for a more informative marker in a region that includes ITS.
52-54: Please include references.
Material and methods
76-78: This information is repeated in the table.
Results
114-115: Do you refers to the assembled sequences, clarify. This sentence fits better in methods.
123: Must be “one and five” instead “five and one”?
142-143: Not clear. The TIS is not in the middle.
146-148: The information of individual, intraspecies and interspecies variation for NTS, ETS and SRs could be described in a table. It is at your choice. It would be interesting. Also, in the same regard, to include the sequences obtained for ITS and RS as figures. I think they can enrich the manuscript.
Discussion
187: I suggest a dot after Bhachiola algare.
Instead “in rDNA length” maybe is better “in the entire rDNA unit length.
191-192: This line sound odd. Sequences differences are not responsible for variations in length bur indels.
190: In length and sequences?
201: After “…evolution of rDNA units” can you add a reference?
208: “…marine invertebrate” should read “…marine invertebrates”.
210: Change rRNA to rDNA.
218: Delete genes.
220-223: It is not clear in the document in what degree are the SRs different. To perform phylogenetic reconstruction as you say you need some degree of conservation.
224: Delete sequences.
228: At you choice. You can explore a little bit more the data, if realize diversity (π) analisys using the dnaSP software, in ITS, ETS and NTS.
References
233: Change 12,1-14 to 12:13.
Line 294: Change “4:293-298” to “4,293-298”.
Tables
Table 4. Heading. Something is missing. It is not clear.
Figures
Fig. 1. Figure caption. Change “in Perna species” to “in P. viridis and P. canaliculus”.
Fig.2. Figure caption. In “methods and based”, delete “and”.

For the rebuttal letter:
The clones you sequence can serve for secondary structure analysis in ITS, this has been done in other mollusks, then you have other structures to compare. Is at your choice, as is not the main objective, but these kind of analysis can enrich the document.

Experimental design

-

Validity of the findings

-

Additional comments

The article is ok. It is a good job. But I'd like that you exploit more the data. They have a lot of potential.

Reviewer 3 ·

Basic reporting

Overall, the manuscript is improved with respect the previous version. Authors accepted and properly addressed all my remarks. Still, I think that English needs some corrections here and there: I suggest the use of either a professional English editing service or the help of a native speaker colleague.
My only point main concern is that the analysis for species identification has been not more taken into account: that could have been a nice part of the work.

Experimental design

The manuscript fit well the scope of PeerJ; I can't find any fatal flaw and it appears to me scientifically sound.

Validity of the findings

Findings are ok, data appears robust and the their discussion appropriate.

---

## Round 0.3 · Minor Revisions

Please, follow the indications of the reviewer. After that we will accept your manuscritp.

Reviewer 2 ·

Basic reporting

17. I think is more correct “the copy number (singular) of repeat units (plural)”.
38-39. COI and ITS can distinguish populations (in different geographic regions). And ITS can serve to asses intraindividual variation. You can change the sentence to say “…not always can distinguish”, or if you are speaking exclusively in Perna populations clarify.
170. “the individuals of them” sounds odd. I suggest to go back to only “the individuals”.

Table 5. Is sounds better: “The rDNA ...used for comparison purposes in this publication.
Or if these are all the sequences published for eukaryots to this date can be: The rDNA…of eukaryots published to date.
Also, include the subregions in the title: “The rDNA unit length (nrDNA), 18S rRNA gene, ITS1, 5.8S rRNA gene, ITS2, 28S rRNA gene and IGS lengths (bp), and GenBank Accession numbers of…”
It would be good to add the data for the two Perna species at the end of the table.
In the subheadings GenBank Accessions NO. should read Genbank Accession No.
In the subheadings, 18S rRNA should be 18S rDNA or 18S rRNA genes, the same for 5.8S and 28S.
Fig. 2. Even when the sequences are so different, it is better to add the alignment, using dashes for deletions and dots for identities. Also the values for diversity, intra- and interspecific identities in table 4.

Experimental design

-

Validity of the findings

-

Additional comments

The article has been significantly improved. Hovewer there are still some minor points that should be corrected. I would be happy to do a final revision to the document.

---

## Round 0.4 · accepted · Accept

Congratularions. Your paper is now accepted.

Reviewer 2 ·

Basic reporting

-

Experimental design

-

Validity of the findings

-

Additional comments

-